# AR/ER Ratio Correlates with Expression of Proliferation Markers and with Distinct Subset of Breast Tumors

**DOI:** 10.3390/cells9041064

**Published:** 2020-04-24

**Authors:** Nelson Rangel, Milena Rondon-Lagos, Laura Annaratone, Andrés Felipe Aristizábal-Pachon, Paola Cassoni, Anna Sapino, Isabella Castellano

**Affiliations:** 1School of Biological Sciences, Universidad Pedagógica y Tecnológica de Colombia, Tunja 150003, Colombia; 2Department of Medical Sciences, University of Turin, 10126 Turin, Italy; 3Pathology Unit, Candiolo Cancer Institute, FPO-IRCCS, 10060 Candiolo, Italy; 4Departamento de Nutrición y Bioquímica, Facultad de Ciencias, Pontificia Universidad Javeriana, Bogotá 110231, Colombia

**Keywords:** breast cancer, androgen receptor, estrogen receptor, crosstalk of nuclear receptors, proliferation genes

## Abstract

The co-expression of androgen (AR) and estrogen (ER) receptors, in terms of higher AR/ER ratio, has been recently associated with poor outcome in ER-positive (ER+) breast cancer (BC) patients. The aim of this study was to analyze if the biological aggressiveness, underlined in ER+ BC tumors with higher AR/ER ratio, could be due to higher expression of genes related to cell proliferation. On a cohort of 47 ER+ BC patients, the AR/ER ratio was assessed by immunohistochemistry and by mRNA analysis. The expression level of five gene proliferation markers was defined through TaqMan^®^-qPCR assays. Results were validated using 979 BC cases obtained from gene expression public databases. ER+ BC tumors with ratios of AR/ER ≥ 2 have higher expression levels of cellular proliferation genes than tumors with ratios of AR/ER < 2, in both the 47 ER+ BC patients (*P* < 0.001) and in the validation cohort (*P* = 0.005). Moreover, BC cases with ratios of AR/ER ≥ 2 of the validation cohort were mainly assigned to luminal B and HER2-enriched molecular subtypes, typically characterized by higher proliferation and poorer prognosis. These data suggest that joint routine evaluation of AR and ER expression may identify a unique subset of tumors, which show higher levels of cellular proliferation and therefore a more aggressive behavior.

## 1. Introduction

The cytoplasmic ligand-dependent transcription factor androgen receptor (AR) has gained increasing attention as an important marker of breast cancer (BC) biology [1]. Nevertheless, its significance is still unclear, since AR positivity has been associated with different clinical outcomes in BC patients, according to their estrogen receptors (ER) status [2,3,4,5]. Besides, regarding AR positivity in BC subtypes, immunohistochemistry (IHC)-surrogate subtype classification indicates that ER-positive/HER2-negative (ER+/HER2-) cancers express AR more frequently (91%) than ER+/HER2-positive (HER2+) (68%), ER-negative (ER-)/HER2+ (59%) and triple negative BC (TNBC) (32%); however assignment of intrinsic molecular subtypes by gene expression analysis, reveals that AR levels are similar in luminal A, B and HER2-enriched, but lowers in Basal-like subtypes [6,7]. Of note BC subtypes have widely shown to possess significant prognostic differences. These controversial results are in line with in vitro studies displaying that AR may induce divergent proliferative responses in ER-positive (ER+) [8,9] and in ER-negative (ER-) BC cell lines [10]. AR and ER expression may co-localize in both, normal and tumor tissues, providing evidence for a possible crosstalk between signaling pathways of these hormone receptors [9]. Recently, we and others have demonstrated that the AR and ER expression ratio (AR/ER) could be associated with outcome in ER+ BC patients [11,12]. In fact, high AR and low ER protein expression levels (AR/ER ratio ≥ 2) are significantly associated with aggressive clinical features, tamoxifen resistance, as well as poor disease-free and overall survival [11]. Furthermore, from a molecular point of view, ER+ tumors with higher AR protein levels, do not always result as a luminal subtype using PAM50 gene expression classifier [12].

It has been demonstrated that AR has the ability to regulate cell cycle progression of breast tumors, through activation of cell signal transduction pathways (e.g., PI3K/mTOR and MAPK) or even, by direct binding of AR to gene promoters involved in cell proliferation [13,14]. Cell proliferation represents one of the most important biological features contributing to cancer aggressiveness and recurrent disease [15,16], thus the assessment of several gene markers, have been shown to provide higher prognostic information in BC patients [17,18]. Among them, *AURKA* (kinase involved in microtubule formation during chromosome segregation), *CCNB1* (its product forms the maturation-promoting factor—MPF) and *UBE2C* (required for the destruction of mitotic cyclins), encode proteins implicated in the control of mitosis. Other gene markers are *BIRC5*, which encodes negative regulatory proteins that prevent apoptotic cell death and *MKI67*, that encodes a nuclear protein usually increased in proliferating cells. These genes, activated by different signaling pathways, play roles directly related with cell cycle control and their expression is increased in BC cells [19]. Accordingly, different multigene prognostic assays currently use *AURKA, BIRC5, CCNB1, MKI67* and *UBE2C* as markers representing a proliferation gene expression signature [20,21].

Considering these data, the aim of the present study was to analyze the relationship between AR/ER ratio and proliferation markers (*AURKA, BIRC5, CCNB1, MKI67* and *UBE2C)* in BC cases grouped according to their AR/ER ratio levels (<2 vs. ≥2). Validation of this relationship was performed using datasets of BC gene expression profiling retrieved from public databases.

## 2. Materials and Methods

### 2.1. Study Design and Population

We selected 47 ER+ primary invasive BC cases with matched fresh-frozen tissue, who underwent surgery (between 2014 and 2017) at the Breast Unit of the Città della Salute e della Scienza University Hospital of Torino, Turin—Italy. Specimens were under-vacuum seal inside the surgical theater immediately after surgery, and kept at 4 °C until transfer to the pathology laboratory [22]. Once in the pathology lab, from each specimen, at least one sample was taken, embedded in OCT^TM^ (Tissue-Tek^®^-Sakura, Torrance, CA, USA) compound, snap frozen in isopentane and stored at −80 °C. Diagnostic samples were routinely processed to paraffin embedding with an automatic processor (Leica ASP 300, Leica Microsystems, Wetzlar, Germany). For all cases the following clinico-pathological data were collected: age, type of surgery (conservative surgery vs radical mastectomy), tumor size (<20 mm vs. ≥20 mm), histological type, tumor grade, nodal involvement, vascular invasion, ER, progesterone receptor (PgR), Ki67 and human epidermal growth factor receptor 2 (HER2) status. In addition, nine ER- BC cases were studied to establish further comparison between them and ER+ tumors. Nine fresh-frozen samples, from non-neoplastic breast tissues, were used as controls. All BC cases were classified following Saint Gallen Consensus meeting (St. Gallen) and American Society of Clinical Oncology/College of American Pathologists Guideline (ASCO/CAP) recommendations [23,24]. Ethical approval for this study was obtained from the Comittee for human Biospecimen Utilization (Department of Medical Sciences—ChBU). The same ethic institutional review board approved that written consent from the patients was not needed, given the retrospective approach of the project and that the study did not interfere with diagnosis or treatment decisions. All the cases were anonymously recorded and data were accessed anonymously.

### 2.2. Immunohistochemistry (IHC)

Histological review was performed for all cases. The most representative formalin-fixed paraffin-embedded tissue block (FFPE) of each case was selected and new slides were obtained for the assessment by IHC of all prognostic markers and AR. IHC was performed using an automated slide processing platform (Ventana BenchMark AutoStainer, Ventana Medical Systems Inc., Tucson, AZ, USA) with the following primary antibodies: prediluted anti-ERα rabbit monoclonal antibody (SP1, Ventana Medical Systems Inc., Tucson, AZ, USA); prediluted anti-PgR rabbit monoclonal antibody (1E2, Ventana Medical Systems Inc., Tucson, AZ, USA); anti-AR mouse monoclonal antibody (AR441, diluted 1:50, Dako, Glostrup, Denmark); anti-Ki67 mouse monoclonal antibody (MIB1, diluted 1:50, Dako, Glostrup, Denmark). Measurement of HER2 expression was performed by anti-HER2 polyclonal antibody (A0485, diluted 1:800, Dako, Glostrup, Denmark). Fluorescence in situ hybridization (FISH) for assessing *HER2* gene amplification was performed on IHC equivocal cases (score 2+) [25]. Positive and negative controls (omission of the primary antibody and IgG-matched serum) were included for each immunohistochemical run.

The cut-off value for ER and PgR-positivity was set at ≥1%, as suggested by St. Gallen and ASCO Guideline Recommendations [23,24] and the same cut-off was also adopted for AR positivity [3]. HER2 status was classified as negative (score 0, 1+ and 2+ by IHC but not amplified by FISH) or positive (when scored 3+ by IHC or HER2 amplified by FISH) according to guidelines [23,26]. The percentage of Ki67 positive cells was recorded and the cut-off for dichotomizing tumors with low and high proliferative fraction was established at 20% [23,27].

### 2.3. RNA Extraction

From OCT^TM^ (Tissue-Tek^®^-Sakura, Torrance, CA, USA) frozen blocks, 10 cryosections (20 μm thick each section) were cut to extract RNA. The suitability of the material was evaluated on one frozen section stained with hematoxylin and eosin, and extraction was performed from tumor samples with a percentage of neoplastic cells higher than 80%. The 10 cryosections were collected in 1 mL of TRIzol^®^ (Life Technologies, Carlsbad CA, USA) in a 1.5 mL sterile Eppendorf tube, and processed according to the manufacturer’s instructions. RNA pellets were resuspended in DEPC-treated water, and the RNA concentrations were measured with a spectrophotometer (NanoDrop 1000, Thermo Scientific, Wilmington, DE, USA). The RNA samples were stored at −80 °C until further analysis.

### 2.4. Reverse-Transcription PCR (RT-PCR)

One μg (1 μg) of total RNA from each sample was treated with DNase I (1 U/μL; Roche Diagnostics, Mannheim, Germany). To obtain complementary DNA (cDNA), we used the high-capacity reverse transcription kit (Applied Biosystems, Foster City, CA, USA) in a reaction mixture with the following components: 1× reaction buffer, 4 mM dNTPs, 1× random hexamers, 0.5 U/μL RNAse inhibitor, 1.25 U/μL MultiScribe Reverse Transcriptase and DEPC-Treated water. The reaction mixture was incubated at 25 °C for 10 min, at 37 °C for 2 h and finally at 85 °C for 5 min. RNA samples without reverse transcriptase were reverse transcribed and used as negative controls for DNA contamination in PCR analysis. RNA integrity and quality of cDNA synthesis were assessed by conventional PCR of control housekeeping gene *G6PDH*. cDNA samples were amplified using the touchdown-PCR conditions first reported by Korbie and Mattick [28]. Experiments were performed in a 25 μL reaction volume using the following components: PCR reaction buffer (1× final), MgCl2 (1.5 mM final), dNTPs mix (200 μM final), primers (each 0.5 μM final), MyTaq DNA polymerase (1.25 U final, Bioline Reagents Limited, London, UK) and distilled water (dH2O). Primers, designed using the Oligo explorer 1.5 Software, amplified a target of 215 bp (primer sequences: Fw. 5′-GAGGCCGTGTACACCAAGAT-3′ Rev. 5′-AGCAGTGGGGTGAAAATACG-3′). The reactions were performed on a PTC-100 Peltier Thermal Cycler (MJ Research, Inc., MA, USA) and PCR products were separated by electrophoresis on a 2% agarose gel-stained with GelRed^®^ (Biotium, Inc., Fremont, CA, USA).

### 2.5. qPCR Assays

cDNA samples were used to assess the expression of *AR, ESR1*, five cell proliferation gene markers (*AURKA, BIRC5, CCNB1, MKI67* and *UBE2C*) and two control housekeeping genes (*ACTB* and *GAPDH*) through TaqMan®-qPCR assays. Assays consist of a pair of unlabeled PCR primers and a TaqMan probe with a FAM dye label on the 5′ end and minor groove binder (MGB) and nonfluorescent quencher (NFQ) on the 3′ end. TaqMan Gene Expression assays were purchased from Applied Biosystems (Foster City, CA, USA). The assays include: *AR* (Assay ID: Hs00171172_m1); *ESR1* (Assay ID: Hs00174860_m1); *AURKA* (Assay ID: Hs01582072_m1); *BIRC5* (Assay ID: Hs04194392_s1); *CCNB1* (Assay ID: Hs01030099_m1); *MKI67* (Assay ID: Hs01032442_m1); *UBE2C* (Assay ID: Hs00964100_g1); *ACTB* (Assay ID: Hs02786624_g1); *GAPDH* (Assay ID: Hs02786624_g1). The cDNA was characterized using the StepOne Real-Time PCR System (Applied Biosystems. Foster City, CA, USA).

Gene expression differences, as well as, *AR/ER*-qPCR ratio were calculated based on 2^-ΔΔCt^ method. Gene expression was classified as significantly different when the fold-change (FC) expression was at least ±2 and P values lower than 0.05. A gene expression pooled proliferation score (cell proliferation signature—CPS) was calculated, taking the average 2^-ΔΔCt^ across the five cell proliferation gene markers for a particular sample. qPCR experiments were carried out in three biological replicates, and from each biological replicate, at least two technical replicates were also performed.

### 2.6. Validation Cohort

To validate the relationship between AR/ER ratio with cellular proliferation, BC gene expression data were downloaded from NCI Genomic Data Commons (GDC) and are openly available at https://gdc.cancer.gov. A total of 1093 primary BC samples were retrieved. Cases with missing ER/PgR information, neoadjuvant therapy and male BCs were excluded, as well as, those classified by PAM50 classifier as normal-like tumors [29]. Thus, 979 cases were finally selected for validation analysis (Appendix A). The gene expression data were provided with level 3 RSEM counts after upper quartile normalization, to maintain the standardization of different platforms or housekeeping genes. AR/ER ratio was calculated by dividing *AR* gene expression counts by *ESR1* gene expression counts, so ER+ BC samples were classified in two groups (cases with AR/ER ratio < 2 and cases with AR/ER ratio ≥ 2).

From all retrieved samples gene expression levels of individual and pooled genes related with cell proliferation (*AURKA, BIRC5, CCNB1, MKI67* and *UBE2C*) were assessed to establish potential associations with AR/ER ratio groups. For each gene, z-normalized expression value was retrieved, and CPS was calculated as an average of the five genes. Using the same strategy, the genes *AURKb, BUB1B, BUB1, CDK1* and *CHEK1* were independently evaluated as an additional BC cell proliferation signature [30]. BC molecular subtypes were defined on the basis of the PAM50 classifier, a 50-gene subtype predictor developed using microarray and quantitative reverse transcriptase polymerase chain reaction data that incorporates the gene expression–based “intrinsic” subtypes luminal A, luminal B, HER2-enriched, and basal-like [20]. The molecular subtype assignment was performed using the R/Bioconductor package TCGABiolinks v. 2.7.1 [31].

### 2.7. AR-Ness Signature on Validation Cohort

To explore AR activity in the AR/ER ratio groups, we evaluate an AR signature recently reported by Daemen and Manning [32] on the validation cohort. Briefly, expression of 14 and 31 genes related to active or inactive/suppressed AR signaling, respectively, was assessed. Expression of each gene was z-normalized and a score (AR-ness score) was defined as the average z-scored expression of 14 positive signature genes, minus the average z-scored expression of 31 negative signature genes.

### 2.8. Statistical Analyses

Pearson’s Chi square test and Fisher’s exact test were preliminary performed to compare clinical categorical variables and to evaluate potential differences in the variable distribution among groups. Student’s *t*-test was performed to determine significant differences in FC values between groups. Test for median and means (Analysis of Variance-ANOVA) were performed. For more than two groups Tukey HSD post-hoc test was performed. *P*-values < 0.05 (*), <0.01 (**), <0.001 (***) and <0.0001 (****) were considered as statistically significant. All tables and graphs show the mean values of at least three independent experiments. For GDC data, ANOVA, Kruskal Wallis *H* and Mann–Whitney *U* tests were also used to assess the differences in gene expression between AR/ER < 2 and AR/ER ≥ 2 groups. Statistical analyses were performed using SPSS v.21.0 (IBM Corporation) and Prism7 v.7.0a. (GraphPad) statistical software, as well as, R: Language and Environment for Statistical Computing.

## 3. Results

### 3.1. Patients and Tumor Characteristics

Clinical and pathological features of 47 ER+, as well as, nine ER- primary invasive BCs are shown in Table 1. Median age of ER+ patients at diagnosis was 62 years. Most of these tumors (76.6%) were non-special type (IDC-NST), followed by mixed type (10.6%) and invasive lobular carcinomas (ILC) (4.3%). Almost half of the ER+ cases (48.9%) were of grade 3, 63.8% were higher than 2 cm in size, 46.8% had not lymph node involvement at diagnosis and vascular invasion was observed in 85.1% of the primary tumors. Following St. Gallen and ASCO/CAP Guideline recommendations [23,26], positive IHC expression of PgR was observed in 63.8% of the ER+ cases; 23.4% were HER2+, 70.2% had high expression levels of Ki67 (≥20%) and most of these cases were classified as luminal B-Like tumors (74.5%). IHC expression of AR was observed in 41 cases (87.2%). On the other hand, ER- cases preferentially had greater tumor size and histological grade, higher levels of Ki67, all of them were negative for PgR expression and classified as non-luminal-like tumors according to the IHC-surrogate subtype classification.

### 3.2. Assessment of AR/ER Ratio

ER+ cases (47 tumors) were used to assess the AR/ER ratio through IHC and qPCR analysis. For the IHC AR/ER ratio, obtained using the percentage of nuclear staining of each receptor, range was 0 – (Carlsbad, CA, USA) in a 1.5 mL. 50 (mean 4,38). In order to assess the mRNA transcript ratios of AR and ER (qPCR), we calculated 2^−ΔΔCt^ differentials of AR and ER expression for each sample utilizing the within-sample reference genes (average Ct) for normalization purposes. The qPCR AR/ER ratio range was −535.6–61 (mean −24,33). Following previous reports, cases were grouped by the AR/ER ratio cut-off (<2 vs. ≥2) which better differentiate patients by prognosis [11,12]. Ten cases (10/47, 21.3%) showed higher AR/ER ratio (AR/ER ≥ 2) by both, IHC and qPCR methods, which further indicates a significant positive correlation between protein and mRNA expression levels (*P* < 0.0001—Appendix A). The clinico-pathological characteristics of ER+ BC cases are associated with AR/ER ratio cut-off (IHC and qPCR methods) in Table 2. In the descriptive analysis, patients with a higher AR/ER ratio were associated with the overexpression/amplification of the *HER2* gene (HER2+; *P* = 0.01) and also with the IHC-Surrogate subtype (*P* = 0.004), following St. Gallen and ASCO/CAP guideline recommendations [23,24].

### 3.3. Association of AR/ER Ratio with Genes Involved in BC Proliferation

AURKA, BIRC5, CCNB1, MKI67 and UBE2C are well characterized genes involved in proliferation of BC cells [33]. The gene expression pooled proliferation score (CPS) from all these markers, showed that ER+ BC patients with AR/ER ratio ≥ 2 (defined by both IHC and qPCR methods) have significantly higher levels of proliferation (*P* < 0.001) compared to cases with AR/ER ratio < 2, even when only ER+/HER2- cases were taken into account (*P* < 0.05). Furthermore, the score in cases with AR/ER ratio ≥ 2 was similar to that observed in ER- BC patients (*P* > 0.05) (Figure 1A,B), usually characterized for higher levels of proliferation [34]. These differences were not observed when tumors were grouped only by ER% positivity levels, defined by IHC (Figure 1C). In contrast to proliferation gene signature by qPCR, Ki67 tested by IHC was not differentially expressed depending on AR/ER ratio and ER levels (Table 2 and Appendix A). To further address this divergent result, correlation analyses between qPCR-MKI67 levels and Ki67-IHC score were carry out. In analyzed cases as a whole, a significant positive correlation could be observed (*P* < 0.0001), however when analyses were performed according to AR/ER ratio groups, the correlation decreased in cases with ratio < 2 (*P* = 0.012) and it was lost in the AR/ER ≥ 2 group (*P* = 0.730) (Appendix A). The same result was observed when correlation analyses were separately performed between CPS score and Ki67-IHC score (Appendix A).

### 3.4. Association of AR/ER Ratio with Cellular Proliferation in Validation Cohort

A pooled gene expression analysis (CPS) of publicly available data sets was also performed on 979 BC patients with available gene expression and complete clinical data. Clinico-pathological characteristics of these patients according to the datasets are reported in Appendix A. AR/ER ratio analysis was performed on 742 ER+ cases. AR/ER ratio was calculated according to the AR and ER mRNA gene expression; 4.4% of these cases (43/742 tumors) showed ratio values greater than two (AR/ER ≥2). Pooled analysis of AURKA, BIRC5, CCNB1, MKI67 and UBE2C genes demonstrated that in this group, proliferation levels were significantly higher (*P* = 0.005) compared with cases with ratio values lower than two (AR/ER < 2) (Figure 2A). Similar results were observed from the analysis, independently performed, using the additional BC cell proliferation signature (AURKb, BUB1B, BUB1, CDK1 and CHEK1 genes—Figure 2B) (*P* = 0.01).

In order to better stratify ER+ cases of the publicly available gene expression data sets, BC molecular subtypes were defined using the PAM50 classifier and grouped in accordance with AR/ER ratio cut-off (<2 vs. ≥2). Molecular subtype distribution was significantly different (*P* < 0.00001) between the AR/ER ratio ≥ 2 BCs, preferentially assigned to luminal B (48.8%) and HER-2 enriched (25.6%) subtypes, and the AR/ER ratio < 2 BCs mainly classified as luminal A (70.8%) (Figure 3).

### 3.5. AR-Ness Signature Evaluation in Validation Cohort

In order to establish possible AR-driven BC tumors, we applied the AR-ness signature to cases subdivided according to AR/ER ratio. As shown in Figure 4, cases with AR/ER ratio ≥ 2 had a significant higher score (score: 0.476) compared with cases with AR/ER ratio < 2 (score: 0.115). This means that in cases with AR/ER ratio ≥ 2, AR-induced genes were on average expressed (active AR signaling) at high level than genes reflecting of inactive AR signaling. On the other side, and in agreement with Daemen and Manning study [32], ER- cases had very low AR-ness score, generally indicating inactive/negative AR-driven tumors.

## 4. Discussion

In this study, we have shown that BC cases with AR/ER ratio ≥ 2 (assessed by IHC and qPCR) are characterized by an increased proliferation status defined by a pooled gene expression analysis of *AURKA, BIRC5, CCNB1, MKI67* and *UBE2C* proliferation markers. In addition, these carcinomas preferentially pertain to Luminal B and HER2-enriched molecular subtypes, suggesting their more aggressive behavior in comparison with the majority of ER+ BC cases.

The role of AR in BC is not totally clear. However, several reports indicate that AR is involved in the regulation of cell proliferation. In particular, high AR expression levels can drive tumor proliferation of ER- BC cell lines [35,36], whereas in ER+ BC cell lines AR expression has been usually associated with decreased cell proliferation [8,9,37,38,39]. This dual role of AR, seems to be mainly explained by the ability of AR to recognize not only androgen response elements (ARE), but also, estrogen response elements (ERE) [9]. Consequently, AR compete with ER to regulate transcription of its target genes, as in classic luminal tumors [1,40]. Interestingly, experiments in ER- BC cell lines, where AR was specifically silenced or inhibited, showed a reduction in mRNA levels of those classic ER target genes, accompanied with decreased cell proliferation [41].

On the whole, the above evidence allows hypothesizing that AR could promote expression of ER target genes related with proliferation, by replacing ER when its levels are low or absent. The data obtained in the present paper are in line with this hypothesis, since luminal tumors (ER+) carrying AR/ER ratio ≥ 2 had higher cell proliferation gene expression (Figure 1A and Figure 2). Besides, our results support recent reports indicating that BC cases with AR/ER ratio ≥ 2 have both shorter disease-free interval (DFI) and overall survival (OS) [11,12]. The differences in proliferation observed between the two groups are not dependent on variable ER expression levels [42,43], because cases divided by ER% positivity alone did not show significant differences in the cellular proliferation signature (Figure 1B). Of note, differences were not observed in the expression of the main IHC proliferation marker—Ki67 among groups. In general, there was a clear positive correlation among methodological approaches used to assess expression of this marker, however specifically in the AR/ER ≥ 2 group, the correlation was not kept (Appendix A). This suggests that single Ki67-IHC evaluation is not sensitive enough to identify increased proliferation rates observed in this BC subset. The above can probably be explained due to a particular biological behavior of this type of tumor and by the intrinsic low sensitivity of a single IHC marker compared to a multigene panel evaluated by highly sensitive qPCR assays.

Several studies suggest that high AR expression may contribute to cell proliferation and be detrimental in luminal BCs. A recent report proved that accelerated degradation of AR promoted growth arrest and apoptosis, not only in ER-, but also in MCF7 (Luminal) cells, and these effects were inverted with reintroduction or overexpression of AR [34]. Additional studies showed that the AR-inhibitor, enzalutamide, abrogate both the androgen and estrogen-induced proliferation of MCF-7 cells, in vitro, and in MCF-7 xenograft models, in vivo, by inhibiting receptors translocation [11]. Moreover, De Amicis et al. [44], demonstrated that Tamoxifen-resistant tumors express higher levels of AR with lower levels of ER. The same study also indicated that Tamoxifen stimulates proliferation and xenograft growth of AR-overexpressing MCF-7 cells. These data are in agreement with results obtained from clinical specimens in which AR/ER ≥ 2 was associated with increased risk of tamoxifen resistance [11]. Finally, emerging evidence in ER+ BC cell lines suggest that high AR expression may also facilitate cell migration because induce CXCL12-CXCR4 expression [45] and downregulation of E-cadherin [46], both events leading to epithelial-to-mesenchymal transition (EMT) and metastasis. Altogether, these results suggest the effect of AR as a tumor promoting factor in ER+ BC cells.

Further, to confirm differences in proliferation, bioinformatics analyses on validation cohorts enabled to determine the molecular subtypes observed in each AR/ER ratio group. Most of the cases with AR/ER ratio ≥ 2 were assigned to luminal categories (66.7%), however 47.1% of them were classified as luminal B tumors (Figure 3), characterized by worse prognosis and perceived as a distinct entity compared with Luminal A subtype [47]. In addition, our analyses showed that almost 30% of cases with AR/ER ratio ≥ 2 were classified as HER2-enriched subtype, which suggest AR and HER2 signaling pathways may collaborate to regulate HER2-enriched tumors [32]. However, it is important to note that differences in proliferation are not only explained by higher HER2 expression/amplification, since ER+/HER2- cases with AR/ER ratio ≥ 2 also had higher levels of cellular proliferation markers (Figure 1C). Although functional crosstalks between AR and HER2 have been described mainly in ER- cells [48], genomic AREs could cause rapid initiation of cytoplasmic signaling cascades (non-genomic mechanisms), like the ErbB (HER family) and MAPK signaling in ER+ cells too [49]. Since these pathways has been associated not only with higher proliferation but also with TAM resistance [50], the evaluation of AR expression in breast tumors, could help to detect a subset of cases possibly AR-driven, as suggested by the AR-ness signature on cases with AR/ER ratio ≥ 2 (Figure 4), which would benefit from anti-AR tailored therapies.

The main limitation of this work was the low number of samples, however analysis on the validation cohort with big number of patients from public databases allowed us to consolidate our data. Although age-related differences were not found in the cases studied (Italian cohort), significant variations, in this regard, were observed in the validation cohort. This contrasting data could be explained, not only, by differences in the number of cases analyzed, but also by differences in the racial/ethnic background composing each of the study populations. Hence, further studies including larger cohorts of patients are needed to figure out these issues and to verify our results.

## 5. Conclusions

In line with previous reports which indicate that BC cases with AR/ER ratio ≥ 2 are characterized to have worse survival, we found that tumors in this group also have higher expression levels of cellular proliferation genes. This was further validated by molecular subtyping using PAM50, since cases with AR/ER ratio ≥ 2 were mainly assigned to luminal B and HER2-enriched subtypes, characterized for higher proliferation and poorer prognoses. Although it has been demonstrated that AR expression is variable across subtypes and that its expression cannot be used as an independent biomarker, our analysis shows that joint routine evaluation of AR and ER expression, should be considered to identify a unique subset of tumors which potentially would benefit from combination of classical and anti-AR based therapies.

## Figures and Tables

**Figure 1 cells-09-01064-f001:**
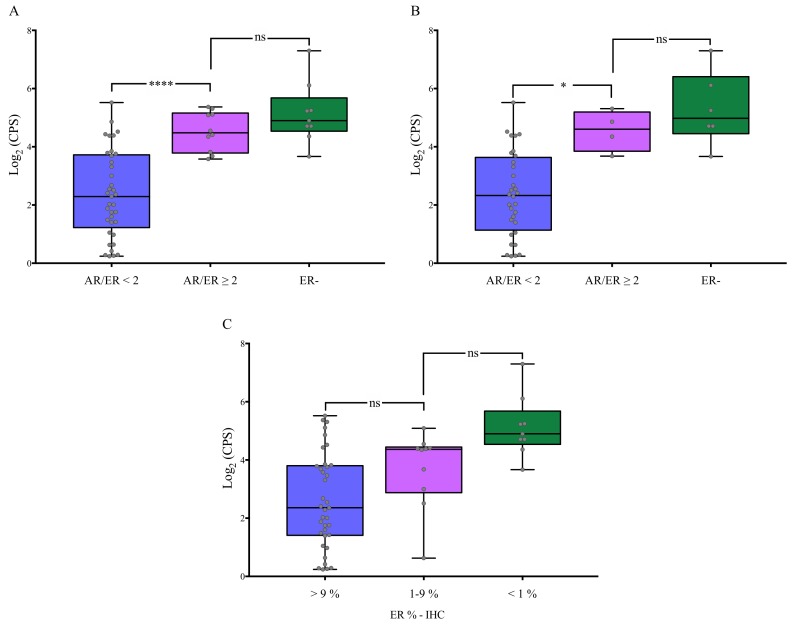
Gene expression cell proliferation signature (CPS) of BC cases grouped by AR/ER ratio, defined by IHC and qPCR methods. Significantly higher levels of CPS were observed in BC cases with ratio AR/ER greater than 2, in both, ER+ cases independently of HER2 status (**A**) and also in ER+/HER2- cases (**B**). However, differences were not identified when tumors were divided by ER% positivity levels (**C**). ER-: estrogen receptor negative <1%. IHC: immunohistochemistry. Tukey‘s multiple comparison test - **** *P* < 0.001; * *P* < 0.05; ns: Not significant.

**Figure 2 cells-09-01064-f002:**
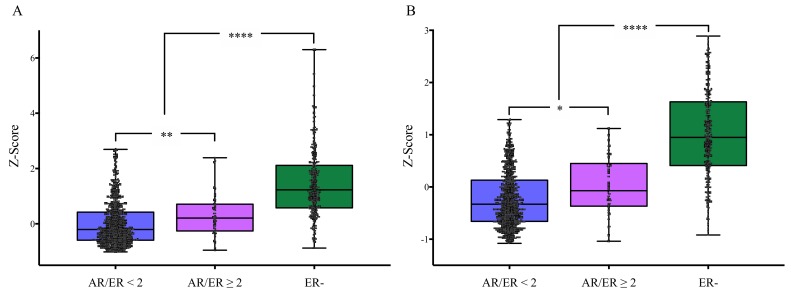
Gene expression cell proliferation signatures (CPSs) of BC cases from GDC, grouped by AR/ER ratio. (**A**) Significantly higher levels of CPS were observed in BC cases with ratio AR/ER greater than 2, using AURKA, BIRC5, CCNB1, MKI67 and UBE2C as gene proliferation markers. (**B**) The same result was observed when AURKb, BUB1B, BUB1, CDK1 and CHEK1 were used, in an independently way, as additional genes to evaluated CPS. A.** *P* = 0.005211; **** *P* = 1.17 × 10^−44^; B.* *P* = 0.028; **** *P* = 3.54 × 10^−54^; ER-: estrogen receptor negative cases; Kruskal–Wallis H test - *, **, ****.

**Figure 3 cells-09-01064-f003:**
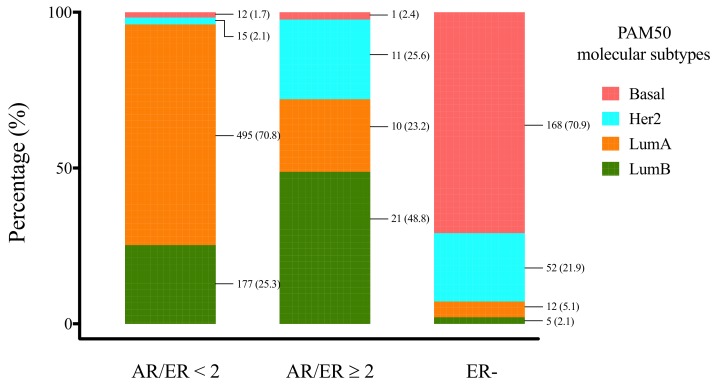
Molecular subtype distribution of BC cases from GDC, grouped by AR/ER ratio. Cases with ratio AR/ER ≥2 were distributed significantly different respect other groups and are mainly represented by luminal B and HER2-enriched tumors. AR/ER ≥ 2 vs. AR/ER < 2: *P* < 0.00001*. ER− vs. Other groups: *P* < 0.00001*. Numbers linked to colours in each bar correspond to total number of cases and its percentage, N (%). ER−: estrogen receptor negative cases. * Chi-Square test (X^2^).

**Figure 4 cells-09-01064-f004:**
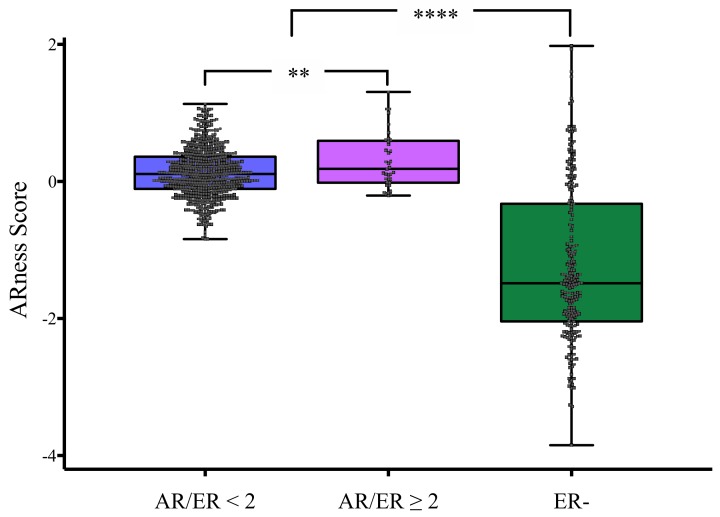
AR-ness signature in BC cases from GDC, grouped by AR/ER ratio. Higher AR-ness score in AR/ER ≥ 2 group indicates a more active AR signaling respect to cases in other groups. ER-: estrogen receptor negative cases. Kruskal–Wallis H test - ** *P* = 0.004, **** *P* = 8.65 × 10^−28^.

**Table 1 cells-09-01064-t001:** Clinical and histopathological characteristics of breast cancer (BC) patients studied. Prognostic marker classification and IHC-surrogate subtype definition were established following St. Gallen and ASCO/CAP guideline recommendations.

Characteristics		ER + n (%)	ER − n (%)	*P* Value (Fisher Test)
Total number of patients		47	9	-
Median age (Interval)		62 (35–82)	60 (43–78)	0.591 *
Tumor Size (missing 1 case)	<20 mm	17 (36.2)	-	0.028
	≥20 mm	30 (63.8)	9 (100)	
Metastatic Lymph nodes	pN0	22 (46.8)	7 (77.8)	0.230
	pN1–3	14 (29.8)	1 (11.1)	
	pN > 3	11 (23.4)	1 (11.1)	
Grading	1	4 (8.5)	-	0.085
	2	20 (42.6)	1 (11.1)	
	3	23 (48.9)	8 (88.9)	
Histotype	IDC-NST	36 (76.6)	5 (55.6)	0.245
	ILC	2 (4.3)	-	
	Mixed type	5 (10.6)	-	
	others	4 (8.5)	4 (44.4)	
Vascular invasion	No	7 (14.9)	3 (33.3)	0.192
	Yes	40 (85.1)	6 (66.7)	
PgR	0	17 (36.2)	9 (100)	<0.0001
	≥1	30 (63.8)	-	
Ki67	<20%	14 (29.8)	1 (11.1)	0.235
	≥20%	33 (70.2)	8 (88.9)	
HER2	Negative	36 (76.6)	6 (66.7)	0.399
	Positive	11 (23.4)	3 (33.3)	
AR	0	6 (12.8)	3 (33.3)	0.148
	≥1%	41 (87.2)	6 (66.7)	
IHC – Surrogate Subtype	Luminal A-Like	12 (25.5)	-	<0.0001
	Luminal B-Like (HER2-)	24 (51.1)	-	
	Luminal B-Like (HER2+)	11 (23.4)	-	
	HER2+/ER-	-	3 (33.3)	
	TNBC	-	6 (66.7)	

DC-NST (invasive ductal carcinom—non special type), ILC (invasive lobular carcinoma). * Student’s *t*-test.

**Table 2 cells-09-01064-t002:** Association of AR/ER ratio with clinico-pathological characteristics of ER+ cases. Classification by AR/ER ratio of ER+ cases was concordant using both IHC and qPCR methods. AR: androgen receptor; ER: estrogen receptor.

Characteristics		AR/ER < 2 n (%)	AR/ER > 2 n (%)	*P* Value (Fisher Test)
Total number of patients		37 (78.7)	10(21.3)	-
Median Age (Interval)		62 (35–79)	65 (47–82)	0.309 *
Grading	1	3 (8.1)	1 (10)	0.215
	2	18 (48.6)	2 (20)	
	3	43 (16.1)	7 (70)	
Tumor size	<20 mm	12 (33.3)	4 (40)	0.485
	≥20 mm	24 (66.7)	6 (60)	
Metastatic Lymph nodes	0	20 (54.1)	2 (20)	0.108
	1–3	9 (24.3)	5 (50)	
	>3	8 (21.6)	3 (30)	
Vascular invasion	No	4 (10.8)	3 (30)	0.155
	Yes	33 (89.2)	7 (70)	
PgR	<20%	26 (70.3)	4 (40)	0.136
	≥20%	11 (29.7)	6 (60)	
Ki-67	<20%	12 (32.4)	2 (20)	0.366
	≥20%	25 (67.6)	8 (80)	
HER2 Status	Negative	32 (86.5)	4 (40)	0.01
	Positive	5 (13.5)	6 (60)	
ER%	Mean (interval)	78 (2–99)	21 (2–50)	<0.001 *
AR%	Mean (interval)	70 (0–99)	83 (50–100)	0.319 *
IHC – Surrogate Subtype	Luminal A	12 (32.4)	0 (0)	0.004
	Luminal B-Like (HER2-)	20 (54.1)	4 (40)	
	Luminal B-Like (HER2+)	5 (13.5)	6 (60)	

* Student’s *t*-test.

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
