# Peer review of "AR/ER Ratio Correlates with Expression of Proliferation Markers and with Distinct Subset of Breast Tumors"

_cells, 2020, doi:10.3390/cells9041064_

Round 1
Reviewer 1 Report
- Article focus: AR/ER ratio >2. vs. <2 prognostic indicator of BC aggression is very interesting. The concept is not entirely novel other groups have suggested that the AR and ER receptor ratio play a factor in BC aggression. This article is among the first to use 2 as a baseline.
- Writing – some slight grammatical issues need to be corrected.
- Figures are not clear please clarify how figure 1A and 1B differ.
- PAM50 classifier group should be defined.
- Figure 3….x-axis needs to be corrected – not sure what AR 2 3
- Overall, well written with interesting findings. The authors need to expound on the following:
- AR, ER, and BC subtypes.
- Proliferation genes and/or pathways
- Racial/ethnic and/or age-related differences between validating cohort and Italian study cohort should be discussed.
- Cell proliferation signature(CPS) -why those five or six genes are considered proliferative markers should be better explained
Author Response
Manuscript ID: cells-755974 – reviewers' comments on manuscript “AR/ER ratio and gene proliferation markers in ER-positive breast cancer patients”
RESPONSES TO REVIEWER 1
Article focus: AR/ER ratio >2. vs. <2 prognostic indicator of BC aggression is very interesting. The concept is not entirely novel other groups have suggested that the AR and ER receptor ratio play a factor in BC aggression. This article is among the first to use 2 as a baseline. Overall, well written with interesting findings.
AUTHORS: We thank the reviewer for considering the idea and results presented in this manuscript interesting.
The authors need to expound on the following:
- Figures are not clear please clarify how figure 1A and 1B differ.
AUTHORS: We thank the reviewer to highlight this point. Following the reviewer’s comment we have clarified difference between figure 1A and 1B changing legend of Figure 1. Now lines 260-262 read: “…Significantly higher levels of CPS were observed in BC cases with ratio AR/ER greater than 2, in both, ER+ cases independently of HER2 status (A) and also in ER+/HER2- cases (B)…”.
- PAM50 classifier group should be defined.
AUTHORS: As suggested by the reviewer, we have now included additional information regarding the PAM50 classifier. Lines 181-185 now read: “…BC molecular subtypes were defined on the basis of the PAM50 classifier, a 50-gene subtype predictor developed using microarray and quantitative reverse transcriptase polymerase chain reaction data that incorporates the gene expression–based “intrinsic” subtypes luminal A, luminal B, HER2-enriched, and basal-like [20]. The molecular subtype assignment was performed using the R/Bioconductor package TCGABiolinks v. 2.7.1 [31]….”.
- Figure 3….x-axis needs to be corrected – not sure what AR 2 3
AUTHORS: Following the reviewer’s comment we checked information on figure 3. On x-axis are found ER+ cases from our validation cohort, grouped by AR / ER ratio (AR/ER≥2 Vs. AR/ER<2), as well as, ER-Negative (ER-) cases and how they are classified according to the molecular subtypes assigned by the PAM50 classifier. Notwithstanding, we have added some clarification regarding numbers next to bars. Now lines 295-296, on legend of figure 3 read: “…Numbers linked to colours in each bar correspond to total number of cases and its percentage, N(%)...”. We hope this clarification is enough to resolve the reviewer's doubt.
- The authors need to expound on AR, ER, and BC subtypes.
AUTHORS: Following reviewer’s suggestion, information about AR and ER markers and its relationship with BC subtypes, have been added in lines 41-47 as follow: “…Besides, regarding AR positivity in BC subtypes, immunohistochemistry (IHC)-surrogate subtype classification indicates that ER-positive/HER2-negative (ER+/HER2-) cancers express AR more frequently (91%) than ER+/HER2-positive (HER2+) (68%), ER-negative (ER-)/HER2+ (59%) and Triple negative BC - TNBC (32%); however assignment of intrinsic molecular subtypes by gene expression analysis, reveals that AR levels are similar in Luminal A, B and HER2-enriched, but lowers in Basal-like subtype [6,7]. Of note BC subtypes have widely shown to possess significant prognostic differences...”.
New references used to enrich this topic [6,7] were included and all references section adjusted.
- The authors need to expound on Proliferation genes and/or pathways
AUTHORS: We appreciate reviewer’s observation as it helps increase clarity of the manuscript. Thus, we have change and added information in lines 62-70. Now these lines read: “…aggressiveness and recurrent disease [15,16], thus the assessment of several gene markers, have been shown to provide higher prognostic information in BC patients [17,18]. Among them, AURKA (kinase involved in microtubule formation during chromosome segregation), CCNB1 (its product forms the maturation-promoting factor - MPF) and UBE2C (required for the destruction of mitotic cyclins), encode proteins implicated in the control of mitosis. Other gene markers are BIRC5, which encodes negative regulatory proteins that prevent apoptotic cell death and MKI67, that encodes a nuclear protein usually increased in proliferating cells. These genes, activated by different signaling pathways, play roles directly related with cell cycle control and their expression is increased in BC cells [19]. Accordingly, different multigene prognostic assays currently use AURKA, BIRC5, CCNB1,…”
New reference used to enrich this topic [19] was included and all references section adjusted.
- The authors need to expound on racial/ethnic and/or age-related differences between validating cohort and Italian study cohort should be discussed.
AUTHORS: We thank the reviewer and we agree with the need to expound this aspect. These issues have been further discussed in lines 376-380 as follows: “…Although age-related differences were not found in the cases studied (Italian cohort), significant variations, in this regard, were observed in the validation cohort. This contrasting data could be explained, not only, by differences in the number of cases analyzed, but also by differences in the racial/ethnic background composing each of the study populations. Hence, further studies including larger cohorts of patients are needed to figure out these issues and to verify our results...”.
- The authors need to expound on Cell proliferation signature (CPS) - why those five or six genes are considered proliferative markers should be better explained
AUTHORS: This is indeed an important issue with respect to the results that we report. The genes used to build the Cell proliferation signature (CPS) were AURKA, BIRC5, CCNB1, MKI67 and UBE2C. To better explain why these five genes are considered proliferative markers we have added information regarding their functions on introduction section; lines 62-70.
REFERENCES RELATED TO RESPONSES
[20] Parker, J.S.; Mullins, M.; Cheang, M.C.; Leung, S.; Voduc, D.; Vickery, T.; Davies, S.; Fauron, C.; He, X.; Hu, Z., et al. Supervised risk predictor of breast cancer based on intrinsic subtypes. J Clin Oncol 2009, 27, 1160-1167, doi:10.1200/JCO.2008.18.1370.
[31] Colaprico, A.; Silva, T.C.; Olsen, C.; Garofano, L.; Cava, C.; Garolini, D.; Sabedot, T.S.; Malta, T.M.; Pagnotta, S.M.; Castiglioni, I., et al. TCGAbiolinks: an R/Bioconductor package for integrative analysis of TCGA data. Nucleic Acids Res 2016, 44, e71, doi:10.1093/nar/gkv1507.
[6] Ringner, M.; Fredlund, E.; Hakkinen, J.; Borg, A.; Staaf, J. GOBO: gene expression-based outcome for breast cancer online. PLoS One 2011, 6, e17911, doi:10.1371/journal.pone.0017911.
[7] Collins, L.C.; Cole, K.S.; Marotti, J.D.; Hu, R.; Schnitt, S.J.; Tamimi, R.M. Androgen receptor expression in breast cancer in relation to molecular phenotype: results from the Nurses' Health Study. Mod Pathol 2011, 24, 924-931, doi:10.1038/modpathol.2011.54.
[15] Gingras, I.; Desmedt, C.; Ignatiadis, M.; Sotiriou, C. CCR 20th Anniversary Commentary: Gene-Expression Signature in Breast Cancer--Where Did It Start and Where Are We Now? Clin Cancer Res 2015, 21, 4743-4746, doi:10.1158/1078-0432.CCR-14-3127.
[16] Stover, D.G.; Coloff, J.L.; Barry, W.T.; Brugge, J.S.; Winer, E.P.; Selfors, L.M. The Role of Proliferation in Determining Response to Neoadjuvant Chemotherapy in Breast Cancer: A Gene Expression-Based Meta-Analysis. Clin Cancer Res 2016, 22, 6039-6050, doi:10.1158/1078-0432.CCR-16-0471.
[17] Dedic Plavetic, N.; Jakic-Razumovic, J.; Kulic, A.; Vrbanec, D. Prognostic value of proliferation markers expression in breast cancer. Med Oncol 2013, 30, 523, doi:10.1007/s12032-013-0523-x.
[18] Inoue, K.; Fry, E.A. Novel Molecular Markers for Breast Cancer. Biomark Cancer 2016, 8, 25-42, doi:10.4137/BIC.S38394.
[19] Dai, X.; Li, T.; Bai, Z.; Yang, Y.; Liu, X.; Zhan, J.; Shi, B. Breast cancer intrinsic subtype classification, clinical use and future trends. Am J Cancer Res 2015, 5, 2929-2943.
Reviewer 2 Report
The data presented very well.
Author Response
Manuscript ID: cells-755974 – reviewers' comments on manuscript “AR/ER ratio and gene proliferation markers in ER-positive breast cancer patients”
RESPONSES TO REVIEWER 2
The data presented very well.
AUTHORS: We thank the reviewer to find the manuscript very well presented.
In order to present clearer information, we have added some modification to the manuscript, following recommendation from other reviewers. You will find all these amendments highlighted in blue colour in the manuscript file.
Reviewer 3 Report
The manuscript "AR/ER ratio and gene proliferation markers in ER-positive breast cancer patients" is a follow-up of the authors on their study showing that AR/ER expression ratio could be useful to predict outcome in the ER+ breast cancer cases. In the current manuscript they provide additional insight in how this ratio is correlating with expression of markers of the cell proliferation signature set and that its increase points toward the HER2+ and Luminal B subtypes. These results are additionally reinforced by data from TCGA. Overall the study is quite well planned and described, however there are some critical points which authors should address:
- the title of the manuscript does not convey the message well, I suggest it should be rephrased, for example as "AR/ER ratio correlates with proliferation markers and points towards distinct subset of tumors"
- authors use a subset of proliferation signature genes (AURKA, BIRC5, CCNB1, MKI67 and UBE2C) and mention that further 5 genes were used as "additional genes to evaluated CPS". This is not clear, since if authors just increased gene set to 10 genes, the remaining 5 original genes will introduce a significant bias. Authors should clearly express that set of other 5 genes was used independently (if it was)
- the separation at AR/ER=2 is overall so far looks rather arbitrary, though no doubt useful. Maybe authors could use this opportunity to find a more data-driven cutoff, e.g. by comparing distributions of AR/ER ratio for Luminal A, B and ER+/HER+ tumors and validating their cutoff value?
- an absence of correlation observed based on Ki67 IHC staining is very surprising, especially given that authors have MKI67 among genes they picked to calculate CPS. They should certainly address this issue better, since it is possible that it depends on the cutoff for Ki67. It would be useful to show and analyze correlation between CPS score and Ki67 IHC score and separately the same between qPCR MKI67 levels and Ki67 IHC.
- the way CPS was calculated from TCGA data is not clearly written in Methods section
Author Response
Manuscript ID: cells-755974 – reviewers' comments on manuscript “AR/ER ratio and gene proliferation markers in ER-positive breast cancer patients”
RESPONSES TO REVIEWER 3
The manuscript "AR/ER ratio and gene proliferation markers in ER-positive breast cancer patients" is a follow-up of the authors on their study showing that AR/ER expression ratio could be useful to predict outcome in the ER+ breast cancer cases. In the current manuscript they provide additional insight in how this ratio is correlating with expression of markers of the cell proliferation signature set and that its increase points toward the HER2+ and Luminal B subtypes. These results are additionally reinforced by data from TCGA. Overall the study is quite well planned and described,
AUTHORS: We thank the reviewer for considering the study well designed and presented.
however there are some critical points which authors should address:
- The title of the manuscript does not convey the message well, I suggest it should be rephrased, for example as "AR/ER ratio correlates with proliferation markers and points towards distinct subset of tumors"
AUTHORS: We greatly appreciate the reviewer's suggestion. We believe that the title option, indicated by reviewer, better reflect the conclusion of this study. Thereby, we have decided to change the title from AR/ER ratio and gene proliferation markers in ER-positive breast cancer patients to "…AR/ER ratio correlates with expression of proliferation markers and with distinct subset of breast tumors…". Changes were made in lines: 2-4
- Authors use a subset of proliferation signature genes (AURKA, BIRC5, CCNB1, MKI67 and UBE2C) and mention that further 5 genes were used as "additional genes to evaluated CPS". This is not clear, since if authors just increased gene set to 10 genes, the remaining 5 original genes will introduce a significant bias. Authors should clearly express that set of other 5 genes was used independently (if it was)
AUTHORS: The reviewer is correct highlighting the need to clearly express that 5 additional genes were independently used to valide the CPS. In order to clarify this point the word “independently” was added in line 180. Thus, lines 179-181 now read: “…Using the same strategy, the genes AURKb, BUB1B, BUB1, CDK1 and CHEK1 were independently evaluated as an additional BC cell proliferation signature…”.
This aspect was also clarified in section results, lines 275-277, as follow: “…Similar results were observed from the analysis, independently performed, using the additional BC cell proliferation signature (AURKb, BUB1B, BUB1, CDK1 and CHEK1 genes – Figure 2B) (P=0.01)…”, and in legend of figure 2, lines 281-282, as follow: “…The same result was observed when AURKb, BUB1B, BUB1, CDK1 and CHEK1 were used, in an independently way, as additional genes to evaluated CPS…”.
- The separation at AR/ER=2 is overall so far looks rather arbitrary, though no doubt useful. Maybe authors could use this opportunity to find a more data-driven cutoff, e.g. by comparing distributions of AR/ER ratio for Luminal A, B and ER+/HER+ tumors and validating their cutoff value?
AUTHORS: We thanks the reviewer for rightly suggested us address this insight. Indeed, in an attempt to find a more data-driven AR/ER ratio value, able to differentiate BC subtypes, we previously applied ROC curve analyses on qPCR AR/ER ratio distributions of the 47 ER+ BCs. Accordingly, much lower values than 2, proved to be useful for this purpose and AR/ER ratio ≥-6 showed to significatively discriminate Luminal B/HER2+ from Luminal A tumors (AUC:0.82; P=0.008); while a ratio ≥-3 allowed to discriminate Luminal B/HER2+ from Luminal B/HER2- (AUC:0.71; P=0.03), as well as, from ER+/HER2- (Luminal A and B/HER2- grouped) tumors (AUC:0.74; P=0.01).
Unfortunately the ratio cutoffs, mentioned above, could not be validated using information from public databases (TCGA). In our study, subtype assignment for the 47 ER+ cases was performed based on IHC profile (IHC-surrogate subtyping), while subtypes in cases from public databases were obtained using PAM50 gene expression classifier (Intrinsic molecular subtypes). Various studies have shown that compatibility between the IHC-based subtype classification and gene expression-based PAM50 classification is modest with a discordance range between 15-30% (Prat. A et al. 2011; Prat. A et al. 2013; Tramm. T et al 2014; Reis-Filho JS and Pusztai L. 2011). Accordingly, we believed that differences in classification systems used, could explain why ratio values of ≥-6 or ≥-3, defined on the 47 ER+ cases, were not validated on public databases.
Due to these inconsistent results, we decided not including those data in the present manuscript and currently we are working to retrieve and collect a bigger number of BC cases which enable us to perform gene expression analyses and determine intrinsic molecular subtypes (PAM50). Thus we could validate if even lower levels of AR respect to ER are useful to discriminate BC subtypes and reinforce the idea that crosstalk among these hormone receptors can modify behavior of BC cells.
- An absence of correlation observed based on Ki67 IHC staining is very surprising, especially given that authors have MKI67 among genes they picked to calculate CPS. They should certainly address this issue better, since it is possible that it depends on the cutoff for Ki67. It would be useful to show and analyze correlation between CPS score and Ki67 IHC score and separately the same between qPCR MKI67 levels and Ki67 IHC.
AUTHORS: We are very grateful with the reviewer, since these comments have encouraged us to take advantage of our data and clarify this point. For analyses presented in table 2, the reviewer is right to suggest that absence of correlation possibly depends on the cutoff used for Ki67 (≥20, following St. Gallen guideline recommendations). However, this correlation was not found either, when analyses were performed using the specific IHC percentage for each case within study groups (Supplementary Figure 2). To further address this issue and following reviewer’s indication, when additional correlation analyses were performed in all cases studied, a significant positive correlation was found, however when analyses were carried on according to AR/ER ratio groups, the correlation decreased in cases with ratio <2 and it was lost in the AR/ER≥2 group.
These new results are described in the section 3.3. Association of AR/ER Ratio with Genes Involved in BC Proliferation, in lines 252-258 as follow: ”…To further address this divergent result, correlation analyses between qPCR-MKI67 levels and Ki67-IHC score were carry out. In analysed cases as a whole, a significant positive correlation could be observed (P<0.0001), however when analyses were performed according to AR/ER ratio groups, the correlation decreased in cases with ratio <2 (P=0.012) and it was lost in the AR/ER≥2 group (P=0.730) (Supplementary Figure 3). The same result was observed when correlation analyses were separately performed between CPS score and Ki67-IHC score (Supplementary Figure 4)…”.
In addition results were discussed in lines 335-342, as follow: ”…Of note, differences were not observed in the expression of the main IHC proliferation marker - Ki67 among groups. In general, there was a clear positive correlation among methodological approaches used to assess expression of this marker, however specifically in the AR/ER≥2 group, the correlation was not kept (Supplementary Figure 3 and 4). This suggests that single Ki67-IHC evaluation is not sensitive enough to identify increased proliferation rates observed in this BC subset. The above, probably explained due to a particular biological behavior of this type of tumors and by the intrinsic low sensitivity of a single IHC marker compared to a multigene panel evaluated by high sensitive qPCR assays.…”.
Correlation analyses are shown in new additional Supplementary Figures 3 and 4 in the Supplementary material file. This information is also indicated in the supplementary materials section, lines 393-393, as follow: “…Figure S3: Correlations between qPCR-MKI67 levels and Ki67-IHC score, Figure S4: Correlations between CPS score and Ki67-IHC score,…”
- The way CPS was calculated from TCGA data is not clearly written in Methods section
AUTHORS: We thank the reviewer for showing the absence of this topic. Information was included in methods, section 2.6, lines 178-179 as follow: “…For each gene, z-normalized expression value was retrieved, and CPS was calculated as an average of the five genes…”.
REFERENCES RELATED TO RESPONSES
Prat A, Cheang MC, Martin M, Parker JS, Carrasco E, Caballero R, et al. Prognostic significance of progesterone receptor-positive tumor cells within immunohistochemically defined luminal A breast cancer. J Clin Oncol. 2013;31(2):203-9.
Prat A, Perou CM. Deconstructing the molecular portraits of breast cancer. Mol Oncol. 2011;5(1):5-23.
Tramm T, Kyndi M, Myhre S, Nord S, Alsner J, Sorensen FB, et al. Relationship between the prognostic and predictive value of the intrinsic subtypes and a validated gene profile predictive of loco-regional control and benefit from post-mastectomy radiotherapy in patients with high-risk breast cancer. Acta Oncol. 2014; 53(10):1337-46.
Reis-Filho JS, Pusztai L. Gene expression profiling in breast cancer: classification, prognostication, and prediction. Lancet. 2011; 378(9805):1812-23.
Round 2
Reviewer 3 Report
The authors correctly and thoroughly responded to the raised concerns and therefore the manuscript can be recommended for publication in its present form.